# Impact of Data Distribution on Fairness Guarantees in Equitable Deep Learning

## Abstract

Fairness in machine learning is paramount to human society because machine learning systems increasingly influence various aspects of our daily lives, particularly in consequence-critical tasks such as medical diagnosis. Deep learning models for medical diagnosis often exhibit biased performance across diverse demographic groups. Theoretical analyses to understand unfairness in AI-based medical diagnosis systems are still lacking. This work presents a comprehensive theoretical analysis of the impact of disease prevalence and data distributions on the fairness guarantees of deep learning models for medical diagnosis. We formalize the fairness problem, introduce assumptions, and derive fairness error bounds, algorithmic complexity, generalization bounds, convergence rates, and group-specific risk bounds. Our analysis reveals that fairness guarantees are significantly influenced by the differences in disease prevalence rates and data distributions across demographic groups. We prove that considering fairness criteria can lead to better performance than standard supervised learning. Empirical results on diverse datasets, including FairVision, CheXpert, HAM10000 and FairFace, corroborate our theoretical findings, demonstrating the impact of disease prevalence and feature distribution disparities on the equitable performance of deep learning models for tasks such as glaucoma, diabetic retinopathy, age-related macular degeneration, and pleural effusion detection. The code for analysis is publicly available via `https://github.com/anonymous2research/fairness_guarantees`.

## 1 Introduction

Fairness in machine learning has become an increasingly important concern, especially in high-stakes applications such as healthcare, where biased predictions can have severe consequences Rajkomar et al. (2018). In the context of medical tasks, equitable deep learning aims to ensure that predictive models perform equally well across various demographic groups, regardless of factors such as gender, race, and ethnicity Chen et al. (2018). However, achieving fairness guarantees in deep learning models is a challenging task, as it requires careful consideration of the variations in data distribution and the prevalence of the target variable (e.g., disease prevalence) across different groups Oakden-Rayner et al. (2020).

Recent work has focused on developing fairness-aware learning algorithms and analyzing their theoretical properties. For example, Zietlow *et al.* Zietlow et al. (2022) formulate the fairness problem as minimizing the absolute difference in the expected per-group error between two groups. Also, the fairness problem in predictive ability has been formalized as minimizing the difference in the expected loss across all demographic groups Dwork et al. (2012). This definition provides a mathematical framework for studying fairness in machine learning models and has been used to derive various fairness error bounds and convergence guarantees Agarwal et al. (2018); Zafar et al. (2017). However, while existing strategies have improved model fairness to some extent, there has been limited theoretical analysis of fairness learning in the context of medical applications Pfohl et al. (2019). The implications of biases are particularly severe in medical settings, where inaccurate or unfair assessments can directly impact patient outcomes, exacerbate health disparities, and undermine trust in healthcare systems Char et al. (2018); Vyas et al. (2020).

In addition, previous research has investigated the relationship between data distribution, class prevalence, and fairness in machine learning Hajian et al. (2015); Chouldechova (2017). For example, Hajian and Domingo-Ferrer examine how class imbalance affects the fairness of classification models Hajian et al. (2015), revealing that performance can vary significantly across different groups when class distribution is uneven. Similarly, Chouldechova explores the impact of data distribution on the fairness of risk assessment models in the criminal justice system Chouldechova (2017), demonstrating that the choice of training data distribution can substantially influence the fairness properties of the resulting model. Nonetheless, these studies do not address medical tasks, particularly the impact of disease prevalence and data distribution in this context.

Building upon these insights, this work aims to provide a comprehensive theoretical analysis of the impact of disease prevalence and data distribution on the fairness guarantees of equitable deep learning models in medical tasks. By leveraging the fairness problem formulation (Definition 1) and the theoretical results on fairness error bounds (Theorem 2), convergence rates (Theorem 5), and group-specific risk bounds (Theorem 6), we derive new analytical bounds that explicitly account for the heterogeneity in data distributions and disease prevalence across demographic groups. These bounds provide a deeper understanding of the factors that influence the fairness properties of deep learning models and can guide the development of more robust and equitable algorithms for medical applications.

The main contributions of this work are as follows:

- We formalize the fairness problem in terms of minimizing the differences in expected loss across various demographic groups, such as race categories, and introduce assumptions on data distributions and loss functions.
- We derive a series of theorems that establish fairness error bounds, algorithmic complexity, generalization bounds, convergence rates, and group-specific risk bounds under various assumptions on the data distributions and loss functions. These results shed light on the key factors influencing the fairness properties of the learned models and provide insights into the design of more robust and equitable algorithms.
- We prove that under certain conditions, the local optima of the fairness problem can outperform those of the supervised learning problem, highlighting the importance of considering fairness criteria in model development.
- We corroborate our theoretical findings with empirical results on diverse datasets, such as FairVision Luo et al. (2023), CheXpert Irvin et al. (2019) HAM10000 Tschandl et al. (2018) and FairFace Maze et al. (2018), demonstrating the impact of disease prevalence and data distributions on the equitable performance of deep learning models for tasks such as glaucoma screening.

The rest of this paper is organized as follows. In Section 2, we discuss related work on fairness learning and datasets. Section 3 presents our main theoretical results on the impact of disease prevalence and data distribution on fairness guarantees. In Section 4, we discuss the empirical validation of the theoretical findings on FairVision and CheXpert. Section 5 concludes the paper and outlines directions for future research. Additionally, in the Appendix, we provide supplementary theoretical proofs and experimental results on the HAM10000 and FairFace datasets.

## 2 RELATED WORK

### 2.1 FAIRNESS LEARNING

In recent years, machine learning has achieved exceptional performance across various fields, yet it often produces biased predictions against different demographic groups. To address this issue, fairness learning has been proposed to eliminate or reduce discrimination and bias against certain protected groups, ensuring fair treatment across different groups. Existing fairness learning approaches can generally be categorized into three strategies: *pre-processing*, which balances the dataset through sampling Wang et al. (2020); Kamiran & Calders (2012) or generative models Ngxande et al. (2020); *in-processing*, which incorporates constraints into the loss function Xu et al. (2020); and *post-processing*, which transforms the model output to ensure fairness Hardt et al. (2016); Pleiss et al. (2017).

However, while these strategies have improved model fairness to some extent, there has been limited theoretical analysis of fairness learning. Corbett-Davies et al. (2017) demonstrates that the trade-off between improving public safety and satisfying current notions of algorithmic fairness can be achieved through theoretical proof and empirical analysis on Broward County, Florida's crime data. Additionally, Lipton et al. (2018) shows that increasing treatment disparity can enhance impact parity through theoretical analysis on simulated and real-world educational data. Recently, Jang et al. (2024) proposes a novel fair representation learning method, which achieves non-separability in latent distribution w.r.t. sensitive features by regularizing data distribution and increases separability w.r.t. the target label by maximizing the marginal distance of decision boundaries among different classes. In contrast to the prior works, this work offers a more comprehensive and focused analysis in the context of medical applications. Specifically, it focuses on the implications of the theoretical findings on fairness error bounds, algorithmic complexity, generalization bounds, convergence rates, and group-specific risk bounds in medical tasks.

## 2.2 FAIRNESS DATASET

Fairness learning has been extensively studied across various fields, with several fairness datasets being proposed in areas such as finance Asuncion & Newman (2007); Ruggles et al. (2015), criminology Dressel & Farid (2018), social sciences McGinley (2010), and education Miao (2010); Kuzilek et al. (2017). Among these fields, healthcare emerges as a critically high-stakes domain where biased predictions can lead to significant and far-reaching consequences. While there are a number of medical datasets that can be used for fairness learning Irvin et al. (2019); Johnson et al. (2019); Kovalyk et al. (2022); Tschandl et al. (2018); Groh et al. (2021); Zambrano Chaves et al. (2021); Afshar et al. (2021); Farsiu et al. (2014); Wyman et al. (2013), most of these datasets either have small sample sizes up to several hundred patients hindering robust assessment of model fairness or limited demographic attributes of age and gender restricting the dataset's ability for comprehensive fairness studies. So far, there are three large-scale medical datasets particularly suitable for fairness learning. First, the FairVision Luo et al. (2023) dataset includes 30,000 fundus photos from 30,000 patients, covering three major eye diseases: age-related macular degeneration, diabetic retinopathy, and glaucoma. This dataset provides comprehensive identity attributes for each patient, including age, gender, race, ethnicity, preferred language, and marital status, all of which are valuable for fairness learning. Second, the CheXpert Irvin et al. (2019) dataset contains 224,316 chest radiographs of 65,240 patients, labeled for the presence of 14 common chest radiographic observations. This dataset provides attribute labels for each patient's age, gender, and race, which are essential for fairness learning. Third, the HAM10000 dataset consists of 9,948 dermatoscopic images of pigmented skin lesions, offering detailed annotations for seven diagnostic categories and providing metadata for patient age and gender, making it a relevant dataset for studying fairness in skin lesion classification. In addition to these medical datasets, the FairFace dataset, which includes 152,917 facial images from 6,100 unique identities, annotated for protected attributes such as gender and skin color, is commonly used to evaluate fairness in computer vision models beyond the medical domain.

## 3 MAIN RESULTS

In this section, we present our main theoretical results on the impact of disease prevalence and data distribution on the fairness guarantees. We provide a series of theorems that establish fairness error bounds, algorithmic complexity, generalization bounds, convergence rates, and group-specific risk bounds under various assumptions on the data distributions and loss functions. These results shed light on the key factors influencing the fairness properties of the learned models and provide insights into the design of more robust and equitable algorithms. *Due to space limitations, all the theorem proofs are deferred to Appendix.*

A central question in equitable deep learning is how to formalize the notion of fairness in the context of predictive models. Building upon the insights and the fairness problem formulation introduced by Zietlow et al. (2022), we extend the formulation to multiple groups and introduce the fairness problem in predictive ability as minimizing the difference in the expected loss across all demographic groups.

**Definition 1** (Fairness Problem). *Given an image $x$, its corresponding label $y$, a demographic attribute $a \in \{a_1, a_2, \ldots, a_k\}$ (e.g., race attributes such as Asian, Black, and White), a function $f$*

*mapping $x$ to predicted labels $\hat{y}$, and a loss function $\ell$, the fairness problem in predictive ability is defined as minimizing the difference in the expected loss across all demographic groups:*

$$\min_{f(\cdot)} \max_{i,j} \left| \mathbb{E}_{(x,y)\sim\mathcal{D}_{a_i}}[\ell(f(x),y)] - \mathbb{E}_{(x,y)\sim\mathcal{D}_{a_j}}[\ell(f(x),y)] \right| \tag{1}$$

*where $\mathcal{D}_{a_i}$ represents the data distribution for demographic group $a_i$, and $\mathbb{E}_{(x,y)\sim\mathcal{D}_{a_i}}[\cdot]$ denotes the expectation over the data distribution for group $a_i$.*

In real-world applications, we have the optimization objective $\min_{f\in\mathcal{F}} \{L(f(x),y) + \lambda \cdot \max_{i,j} \left| \mathbb{E}_{(x,y)\sim D_{a_i}}[\ell(f(x),y)] - \mathbb{E}_{(x,y)\sim D_{a_j}}[\ell(f(x),y)] \right| \}$, where $L(f(x),y)$ is a task-specific loss. To clearly understand the fairness problem, we focus on the fairness term in this work.

Definition 1 formalizes the fairness problem in predictive ability for a machine learning model. The chain of logic is as follows: Firstly, we have an image $x$, its corresponding true label $y$, and a demographic attribute $a$ which can take on values from $\{a_1, a_2, \ldots, a_k\}$. For example, the demographic attribute could be race, with categories like Asian, Black, and White. Then, we have a function $f$ that maps the image $x$ to a predicted label $\hat{y}$, and a loss function $\ell$ that measures the difference between the predicted label and the true label. Next, the goal is to minimize the maximum absolute difference in the expected loss across all demographic groups. In other words, we want the model to perform equally well (in terms of the loss function) for all demographic groups. The expectation $\mathbb{E}_{(x,y)\sim\mathcal{D}_{a_i}}[\ell(f(x),y)]$ represents the average loss for demographic group $a_i$, where the data (image-label pairs) are sampled from the distribution $\mathcal{D}_{a_i}$ specific to that group. Finally, by minimizing the maximum absolute difference in the expected loss across all pairs of demographic groups, we ensure that the model's predictive ability is as similar as possible across the groups. This helps mitigate potential biases or unfairness in the model's performance.

Compared to the fairness definition Donini et al. (2018),

$|\mathbb{E}_{(x,y)\sim D_a}[\ell(f(x),y)] - \mathbb{E}_{(x,y)\sim D_b}[\ell(f(x),y)]| \leq \max p_a, p_b \cdot |L^{+,a}(f) - L^{+,b}(f)| + \max 1 - p_a, 1 - p_b \cdot |L^{-,a}(f) - L^{-,b}(f)|$

**Theorem 1** (Connection with Conventional Fairness). *Let $f$ be a classifier, $\ell$ be a loss function bounded by $M$, and $D_{a_i}$ be the data distribution for demographic group $a_i$. Let $p_i$ be the proportion of positive samples in group $a_i$. Then, for any pair of groups $a_i$ and $a_j$:*

$$\left| \mathbb{E}_{(x,y)\sim D_{a_i}}[\ell(f(x),y)] - \mathbb{E}_{(x,y)\sim D_{a_j}}[\ell(f(x),y)] \right| \leq \max\{p_i, p_j\} \cdot \left| L^{+,a_i}(f) - L^{+,a_j}(f) \right| + 2M \left| p_i - p_j \right|$$
$$+ \min\{1 - p_i, 1 - p_j\} \cdot \left| L^{-,a_i}(f) - L^{-,a_j}(f) \right|$$

*where $L^{+,a_k}(f) = \mathbb{E}[\ell(f(x),y)|y=1, s=a_k]$ is the risk of the positive labeled samples in group $a_k$, and $L^{-,a_k}(f) = \mathbb{E}[\ell(f(x),y)|y=0, s=a_k]$ is the risk of the negative labeled samples in group $a_k$. $L^{+,a_k}(f)$ and $L^{-,a_k}(f)]$ are defined in Donini et al. (2018).*

Disease prevalence is particularly relevant in the context of medical tasks. In many real-world scenarios, the prevalence of a disease can vary significantly across different demographic groups Zhang et al. (2012); Stein et al. (2021). Accounting for the differences in the prevalence is crucial for developing fair and equitable deep learning models. By assuming bounded loss and explicitly considering the disease prevalence for each demographic group, Assumption 1 allows us to derive more accurate fairness guarantees that reflect the real-world challenges in medical applications.

**Assumption 1** (Bounded Loss and Disease Prevalence). *Let $r_i$ be the disease prevalence for demographic group $a_i$, where $\sum_{i=1}^{k} r_i = 1$. Assume that the loss function $\ell$ is bounded, i.e., $0 \leq \ell(\hat{y},y) \leq M$ for some constant $M > 0$.*

**Theorem 2** (Fairness Error Bound). *Under Assumption 1, given the fairness problem as defined in Definition 1, let $f^*$ be the optimal function that minimizes the maximum absolute difference in the expected loss across all demographic groups. Let $\hat{f}$ be an estimate of $f^*$ based on a finite sample of size $n$. Then, by Hoeffding's inequality, with probability at least $1 - \delta$, the following inequality*

*holds:*

$$\max_{i,j} \left| \mathbb{E}_{(x,y)\sim\mathcal{D}_{a_i}}[\ell(\hat{f}(x),y)] - \mathbb{E}_{(x,y)\sim\mathcal{D}_{a_j}}[\ell(\hat{f}(x),y)] \right| \leq$$

$$\max_{i,j} \left| \mathbb{E}_{(x,y)\sim\mathcal{D}_{a_i}}[\ell(f^*(x),y)] - \mathbb{E}_{(x,y)\sim\mathcal{D}_{a_j}}[\ell(f^*(x),y)] \right| + M\sqrt{\frac{\log(2k/\delta)}{2n\min\{r_i\}}}$$

**Remark**. Theorem 2 provides an upper bound on the fairness error of the estimated function $\hat{f}$ in terms of the optimal fairness error $\Delta(f^*)$ and a term that depends on the sample size $n$, the number of demographic groups $k$, the confidence level $\delta$, and the minimum prevalence $\min_i r_i$. Specifically, the bound suggests that to achieve a smaller fairness error, one should have a larger sample size $n$, a smaller number of demographic groups $k$, a higher confidence level $1 - \delta$, and a more balanced distribution of disease prevalence $r_i$ across the groups. In the context of medical applications, this result highlights the importance of collecting sufficient and diverse data from each demographic group to ensure equitable performance. Moreover, it emphasizes the need to consider the heterogeneity in disease prevalence when designing fair and accurate screening models.

**Assumption 2** (Lipschitz Continuity). *Assume that the loss function $\ell$ is Lipschitz continuous with respect to its first argument, i.e., there exists a constant $L > 0$ such that $|\ell(\hat{y}_1, y) - \ell(\hat{y}_2, y)| \leq L|\hat{y}_1 - \hat{y}_2|$ for all $\hat{y}_1, \hat{y}_2, y$.*

**Theorem 3** (Algorithmic Complexity of Fairness Problem). *Under Assumptions 1 and 2, given the fairness problem as defined in Definition 1 with $k$ demographic groups and a function class $\mathcal{F}$ with finite Vapnik-Chervonenkis (VC) dimension $d$, there exists an algorithm that finds an $\epsilon$-optimal solution $\hat{f}$ to the fairness problem with probability at least $1-\delta$, using $O(\frac{k^2}{\epsilon^2}(d\log(k/\epsilon)+\log(k/\delta)))$ samples and $O(k^2|\mathcal{F}|)$ time complexity.*

**Remark**. Theorem 2 sheds light on the sample and time complexity of finding an $\epsilon$-optimal fair solution. The sample complexity grows quadratically with the number of demographic groups $k$ and inversely with the square of the desired accuracy $\epsilon$, as we need to ensure uniform convergence for all pairs of demographic groups. Similarly, the time complexity increases by a factor of $k^2$ due to the pairwise comparisons of the empirical loss functions. These results highlight the challenges in achieving fairness in large-scale medical applications with multiple demographic groups and complex models.

**Theorem 4** (Fairness Generalization Bound). *Under Assumptions 1, given the fairness problem as defined in Definition 1 with $k$ demographic groups and a function space $\mathcal{F}$ with VC dimension $d_{VC}(\mathcal{F})$, for any $\delta > 0$, with probability at least $1 - \delta$, for all $f \in \mathcal{F}$:*

$$\max_{i,j}|R_i(f) - R_j(f)| \leq \max_{i,j}|R_{emp,i}(f) - R_{emp,j}(f)| + M\sqrt{\frac{8(d_{VC}(\mathcal{F})\ln(2em/d_{VC}(\mathcal{F})) + \ln(4k^2/\delta))}{m}}$$

*where $R_i(f) = \mathbb{E}_{(x,y)\sim\mathcal{D}_{a_i}}[\ell(f(x),y)]$ is the expected risk for group $a_i$, $R_{emp,i}(f) = \frac{1}{m_i}\sum_{j=1}^{m_i}\ell(f(x_j^{(i)}), y_j^{(i)})$ is the empirical risk for group $a_i$, $m_i$ is the sample size for group $a_i$, and $m = \sum_{i=1}^{k} m_i$ is the total sample size.*

**Remark**. Theorem 4 (Fairness Generalization Bound) is a key result that provides an upper bound on the fairness risk of a learned model in terms of its empirical fairness risk, the VC dimension of the function space, the number of demographic groups, and the sample size. This bound is crucial for understanding the generalization performance of fair learning algorithms and the factors that influence their ability to produce equitable models. The theorem suggests that to achieve a smaller fairness risk, one should have a larger sample size, a smaller VC dimension, and a smaller number of demographic groups. These insights are in line with the well-known bias-complexity trade-off in statistical learning theory Shalev-Shwartz & Ben-David (2014), where models with lower complexity (i.e., smaller VC dimension) tend to have better generalization performance.

**Theorem 5** (Convergence of Fairness Risk Minimizer). *Let $\mathcal{F}$ be a function space with VC dimension $d_{VC}(\mathcal{F})$ and let $f^* = \arg\min_{f\in\mathcal{F}} R(f)$ be the fairness risk minimizer. Let $\hat{f}_S = \arg\min_{f\in\mathcal{F}} R_{emp}(f, S)$ be the empirical fairness risk minimizer based on a training set $S$ of size*

*m. Then, under Assumptions 1 and 2, for any $\delta > 0$, with probability at least $1 - \delta$ over the random choice of $S$:*

$$R(\hat{f}_S) - R(f^*) \leq \frac{2LM}{\sqrt{m}} \left( \sqrt{2dVC(\mathcal{F}) \ln \frac{em}{d_{VC}(\mathcal{F})}} + \sqrt{2 \ln \frac{4}{\delta}} \right)$$

**Remark**. Theorem 5 (Convergence of Fairness Risk Minimizer) is a fundamental result that characterizes the convergence rate of the empirical fairness risk minimizer to the true fairness risk minimizer. The theorem shows that the excess fairness risk, defined as the difference between the fairness risk of the empirical minimizer and the true minimizer, converges to zero at a rate of $O(1/\sqrt{m})$, where $m$ is the sample size. The convergence rate has important implications for the sample complexity of fair learning algorithms. It suggests that to achieve a desired level of accuracy, the sample size should grow quadratically with the inverse of the desired accuracy. This sample complexity is higher than that of the standard empirical risk minimization Vapnik (1999), which has a sample complexity of $O(1/\epsilon^2)$ for an accuracy level of $\epsilon$ Bottou & Bousquet (2007). Moreover, as the size of the training set increases, the empirical fairness risk minimizer approaches the true fairness risk minimizer, ensuring the convergence of the learning algorithm to a fair solution.

**Assumption 3** (Normal Distribution for Group $a_i$). *Assume that the data distribution for demographic group $a_i$ follows a normal distribution with mean $\mu_i$ and covariance matrix $\Sigma_i$, i.e., $(x, y) \sim \mathcal{N}(\mu_i, \Sigma_i)$ for $(x, y) \in \mathcal{D}_{a_i}$.*

**Theorem 6** (Group-Specific Risk Bound Theorem for Normal Distributions). *Let $\mathcal{F}$ be a function space with VC dimension $d_{VC}(\mathcal{F})$ and let $f_i^* = \arg\min_{f \in \mathcal{F}} R_i(f)$ be the risk minimizer for demographic group $a_i$. Let $\hat{f}_S = \arg\min_{f \in \mathcal{F}} R_{emp}(f, S)$ be the empirical risk minimizer based on a training set $S$ of size $m$ drawn from the overall data distribution. Then, under Assumptions 2 and 3, for any $\delta > 0$, with probability at least $1 - \delta$ over the random choice of $S$:*

$$R_i(\hat{f}_S) - R_i(f_i^*) \leq \frac{2LM}{\sqrt{m}} \left( \sqrt{2dVC(\mathcal{F}) \ln \frac{em}{dVC(\mathcal{F})}} + \sqrt{2 \ln \frac{4}{\delta}} \right) + L|\mu_i - \mu|_2 + L\sqrt{|\Sigma_i - \Sigma|F}$$

*where $R_i(f) = \mathbb{E}_{(x,y) \sim \mathcal{D}_{a_i}}[\ell(f(x), y)]$ is the expected risk for group $a_i$, $R_{emp}(f, S) = \frac{1}{m} \sum_{j=1}^m \ell(f(x_j), y_j)$ is the empirical risk based on the training set $S$, $\mu$ and $\Sigma$ are the mean and covariance matrix of the overall data distribution, and $|\cdot|_F$ denotes the Frobenius norm.*

**Remark**. Theorem 6 establishes that for a specific demographic group $a_i$ with a normal data distribution, the excess risk of the empirical risk minimizer trained on the overall data distribution, $R_i(\hat{f}_S) - R_i(f_i^*)$, can be bounded by three terms: The first term is the same as in Theorem 5 and depends on the VC dimension of the function space, the Lipschitz constant of the loss function, the boundedness of the loss function, and the size of the training set $S$. The second term is proportional to the Euclidean distance between the means of the group-specific distribution and the overall distribution, $|\mu_i - \mu|_2$. The third term is proportional to the square root of the Frobenius norm of the difference between the covariance matrices of the group-specific distribution and the overall distribution, $\sqrt{|\Sigma_i - \Sigma|_F}$. The result implies that the accuracy of the empirical risk minimizer on group $a_i$ depends not only on the size of the training set and the complexity of the function space but also on how close the group-specific distribution is to the overall distribution in terms of their means and covariance matrices. If the means and covariances are similar, the excess risk will be smaller, indicating better accuracy on group $a_i$. Conversely, if the means and covariances differ significantly, the excess risk will be larger, indicating potential accuracy disparities for group $a_i$.

The first term depends on the VC dimension of the hypothesis space, the Lipschitz constant of the loss function, the boundedness of the loss function, and the size of the training set $S$. The second term is proportional to the Euclidean distance between the means of the group-specific distribution and the overall distribution, $|\mu_i - \mu|_2$. The third term is proportional to the square root of the Frobenius norm of the difference between the covariance matrices of the group-specific distribution and the overall distribution, $\sqrt{|\Sigma_i - \Sigma|_F}$.

The result implies that the accuracy of the empirical risk minimizer on group $a_i$ depends not only on the size of the training set and the complexity of the hypothesis space but also on how close the group-specific distribution is to the overall distribution in terms of their means and covariance

matrices. If the means and covariances are similar, the excess risk will be smaller, indicating better accuracy on group $a_i$. Conversely, if the means and covariances differ significantly, the excess risk will be larger, indicating potential accuracy disparities for group $a_i$.

**Corollary 1** (Fairness-Accuracy Trade-off). *Let $\mathcal{F}$ be a function space with VC dimension $d_{VC}(\mathcal{F})$, and let $f^* = \arg\min_{f \in \mathcal{F}} R(f)$ and $f_i^* = \arg\min_{f \in \mathcal{F}} R_i(f)$ be the fairness and accuracy risk minimizers, respectively. Let $\hat{f}_S = \arg\min_{f \in \mathcal{F}} R_{emp}(f, S)$ be the empirical fairness risk minimizer based on a training set $S$ of size $m$ drawn from the overall data distribution. Then, under Assumptions 1, 2, and 3, for any $\delta > 0$, with probability at least $1 - \delta$ over the random choice of $S$:*

$$R_i(f^*) - R_i(f_i^*) \leq \frac{4LM}{\sqrt{m}} \sqrt{2d_{VC}(\mathcal{F}) \ln \frac{em}{d_{VC}(\mathcal{F})} + 2\ln\frac{4}{\delta}} + L|\mu_i - \mu|_2 + L\sqrt{|\Sigma_i - \Sigma|_F}$$

**Remark**. Corollary 1 quantifies the trade-off between fairness and accuracy in equitable deep learning models. The left-hand side of the inequality, $R_i(f^*) - R_i(f_i^*)$, represents the difference in accuracy between the fairness risk minimizer $f^*$ and the accuracy risk minimizer $f_i^*$ for a specific demographic group $a_i$. The right-hand side provides an upper bound on this difference, which depends on the VC dimension of the function space, the Lipschitz constant of the loss function, the sample size, and the dissimilarity between the group-specific distribution and the overall distribution. This result has important implications for the design and evaluation of fair learning algorithms in medical applications. It suggests that achieving perfect fairness (i.e., $R_i(f^*) = R_i(f_i^*)$ for all groups) may come at the cost of reduced accuracy, especially when the data distributions differ significantly across demographic groups. Practitioners should carefully consider this trade-off when developing equitable models and assess the impact of fairness constraints on the model's performance for each group Hardt et al. (2016). Also, the corollary highlights the importance of collecting representative data from each demographic group to mitigate the accuracy disparities induced by distributional differences. By reducing the dissimilarity between the group-specific distributions and the overall distribution, one can tighten the fairness-accuracy trade-off and achieve more equitable performance across all groups Chen et al. (2018).

**Theorem 7** (Expected Loss Bound for Demographic Group with Normal Distribution). *Let $\mathcal{D}_{a_i} \sim \mathcal{N}(\mu_i, \Sigma_i)$ be the data distribution for demographic group $a_i$, and let $\mathcal{D} \sim \mathcal{N}(\mu, \Sigma)$ be the overall data distribution. Let $f(\cdot)$ be a function that maps input $x$ to predicted labels $\hat{y}$, and let $\ell$ be a loss function bounded by $B$, i.e., $|\ell(f(x), y)| \leq B$ for all $x, y$. Suppose we have a training set $(x_j, y_j)_{j=1}^n$ of size $n$ drawn independently from $\mathcal{D}$. Then, with probability at least $1 - \delta$, the expected loss of $f(\cdot)$ on $\mathcal{D}_{a_i}$ is bounded as follows:*

$$\mathbb{E}_{(x,y) \sim \mathcal{D}_{a_i}}[\ell(f(x), y)] \leq \mathbb{E}_{(x,y) \sim \mathcal{D}}[\ell(f(x), y)] + B|\mu_i - \mu|_2 + B\sqrt{|\Sigma_i - \Sigma|_F}$$

**Remark**. Theorem 7 provides an insight into the performance of the empirical risk minimizer on a specific demographic group when the group's data distribution differs from the overall training distribution. The theorem shows that the excess risk of the empirical risk minimizer on group $a_i$ can be bounded by three terms: (1) the excess risk of the empirical risk minimizer on the overall distribution, (2) the difference in means between the group-specific distribution and the overall distribution, and (3) the difference in covariance matrices between the group-specific distribution and the overall distribution. This result has important implications for understanding the sources of performance disparities across different demographic groups. It suggests that the accuracy of a model on a specific group depends not only on its performance on the overall population but also on how well the group's data distribution matches the overall training distribution. If the means and covariances of the group-specific distribution differ significantly from those of the overall distribution, the model may suffer from poor accuracy on that group Chen et al. (2018). The theorem assumes that the group-specific distribution follows a normal distribution, which is a common assumption in statistical modeling and analysis Murphy (2012). The use of the Euclidean distance between means and the Frobenius norm of the covariance matrix difference allows for a concise and interpretable bound on the excess risk. The insights from Theorem 7 can inform the development of fair and accurate models in medical applications. They highlight the importance of collecting representative data from each demographic group and considering the differences in data distributions when training and evaluating models Buolamwini & Gebru (2018). Additionally, the theorem provides a way to quantify the impact of data distribution mismatch on the model's performance, which can guide the development of targeted data collection and model improvement strategies Kearns et al. (2018).

**Corollary 2** (Correlation between Expected Loss Bound and Feature Distance). *Let $\mathcal{D}_{a_i} \sim \mathcal{N}(\mu_i, \Sigma_i)$ be the data distribution for demographic group $a_i$, and let $\mathcal{D} \sim \mathcal{N}(\mu, \Sigma)$ be the overall data distribution. Let $f(\cdot)$ be a function that maps input $x$ to a discriminative feature $z$ in a metric space, and let $\ell$ be a loss function bounded by $B$, i.e., $|\ell(z, y)| \leq B$ for all $z, y$. Suppose we have a training set $(x_j, y_j)j = 1^n$ of size $n$ drawn independently from $\mathcal{D}$. Let $\bar{z}$ be the centroid of the features generated by $f$ on the overall data distribution, and let $\bar{z}_i$ be the centroid of the features generated by $f$ on the demographic group $a_i$. Then, with probability at least $1 - \delta$, the expected loss of $f(\cdot)$ on $\mathcal{D}_{a_i}$ is bounded as follows:*

$$\mathbb{E}_{(x,y)\sim\mathcal{D}_{a_i}}[\ell(f(x), y)] \leq \mathbb{E}_{(x,y)\sim\mathcal{D}}[\ell(f(x), y)] + B \cdot d(\bar{z}_i, \bar{z}) + B\sqrt{\mathbb{E}_{z\sim f(\mathcal{D}_{a_i})}[d^2(z, \bar{z}_i)] - \mathbb{E}_{z\sim f(\mathcal{D})}[d^2(z, \bar{z})]}$$

*where $d(\cdot, \cdot)$ is a distance function in the metric space.*

**Remark**. Corollary 2 demonstrates that the expected loss bound for a demographic group $a_i$ depends on three terms: (1) the expected loss on the overall data distribution, (2) the distance between the group-specific feature centroid and the overall feature centroid, and (3) the difference in the average squared distances of the features from their respective centroids. This result highlights the importance of learning discriminative features that are well-clustered around their centroids and have similar distributions across different demographic groups Zemel et al. (2013). By minimizing the distance between the group-specific centroids and the overall centroid, as well as reducing the discrepancy in the feature distributions, one can tighten the expected loss bound and achieve more equitable performance.

## 4 EXPERIMENTS

**Datasets**. *FairVision* contains 30,000 2D scanning laser ophthalmoscopy (SLO) fundus images from 30,000 patients, and each patient has six demographic identity attributes available. This dataset features three common ophthalmic diseases: DR, AMD, and Glaucoma, with 10,000 samples for each disease Luo et al. (2023). According to the official configuration, for each disease, 6,000 samples are used as the training set, 1,000 as the validation set, and 3,000 as the test set. We select the race attribute with SLO fundus images as the focus of our study.

*CheXpert* is a large dataset of chest X-rays labeled for 14 common chest pathologies from associated radiologist reports. The dataset provides three demographic identity attributes: age, gender, and race Irvin et al. (2019). Following the split setting in Gichoya et al. (2022); Glocker et al. (2023), we select a total of 42,884 patients with 127,118 chest X-ray scans. Among these, 76,205 are used for the training set, 12,673 for the validation set, and 38,240 for the test set. In our experiments, we investigate the detection of pleural effusion across different race demographic attributes.

*HAM10000*, consisting of 10,015 dermatoscopic images, was collected over a span of 20 years from the Department of Dermatology at the Medical University of Vienna, Austria, and a dermatology practice in Queensland, Australia, providing a diverse and representative collection of pigmented skin lesions. Following Zong et al. (2022), after filtering out images with missing sensitive attributes, we obtained a refined subset of 9,948 images, and grouped the original seven diagnostic labels into two categories: benign and malignant, to simplify the analysis and facilitate binary classification.

*FairFace* is a newly curated dataset consisting of approximately 13,000 images from 3,000 new subjects, combined with a reannotated version of the IJB-C dataset Maze et al. (2018), resulting in a total of 152,917 facial images from approximately 6,100 unique identities. The dataset is divided into three subsets: 100,186 images for training, 17,138 images for validation, and 35,593 images for testing. It is comprehensively annotated for protected attributes such as gender and skin color, as well as additional features including age group, eyeglasses, head pose, image source, and face size. For our study, we focus on glasses detection by formulating it as a binary classification problem that aims to distinguish between images with and without eyeglasses.

**Implementation Details/Training Scheme**. We select two deep learning models with different frameworks as our baseline models: the CNN-based EfficientNet Tan & Le (2019) and the Transformer-based ViT Dosovitskiy et al. (2020). All experiments are conducted on an A100 GPU with 80GB of memory. We initialize both EfficientNet and ViT using pre-trained weights provided by TorchVision. During the fine-tuning phase, EfficientNet is trained for 10 epochs with a learning rate of 1e-4 and a batch size of 10. Similarly, ViT is trained for 10 epochs with a learning rate of 1e-4 and a batch size of 50.

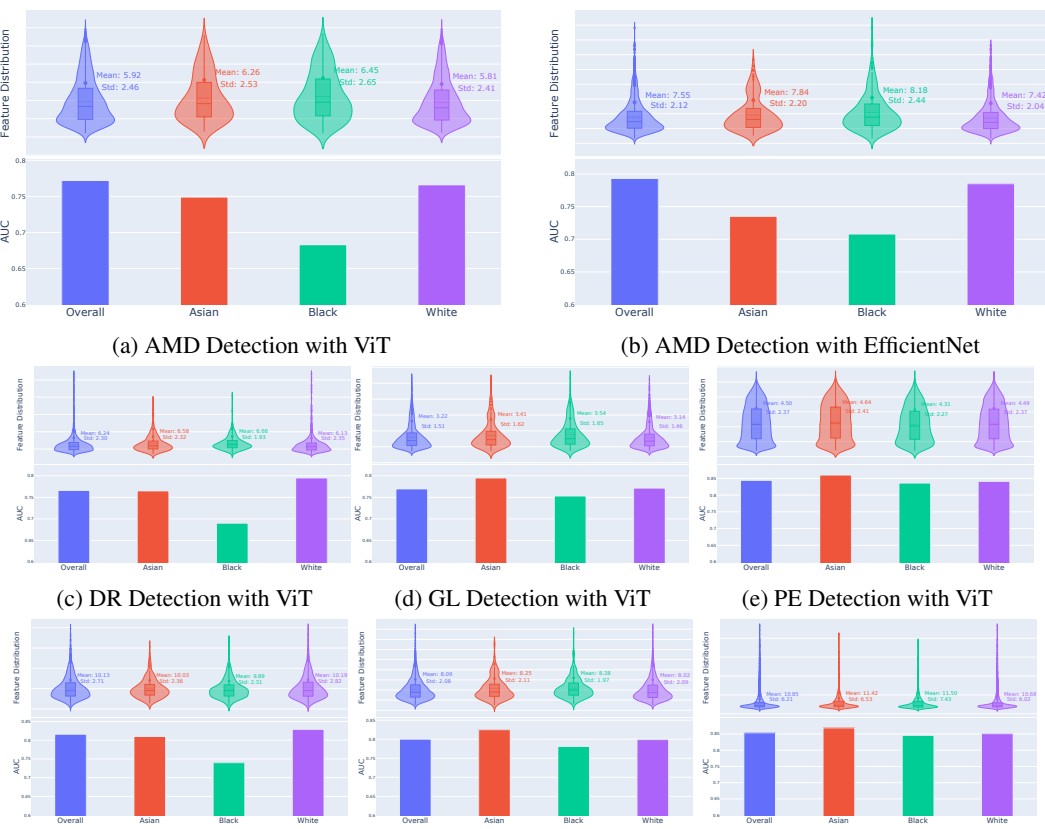

Figure 1: Combination of feature distribution and AUC with ViT and EfficientNet across four demographic groups (Overall, Asian, Black, and White), including AMD, DR, and Glaucoma (GL) detection on FairVision, and pleural effusion (PE) detection on CheXpert.

We train the corresponding EfficientNet and ViT models on the training sets of FairVision for DR, AMD, and Glaucoma, as well as on the training sets of CheXpert, HAM10000, and FairFace. Subsequently, we obtain the features during the encoding phase and the predicted labels. Based on these two outputs, we calculate the Euclidean distance from the mean to represent the empirical distribution of the features, and compute the overall and group-wise AUC accuracy. The feature distributions and AUCs yielded by the two models on the FairVision and CheXpert datasets are shown in Figure 1. For the remaining two datasets, HAM10000 and FairFace, the feature distributions and AUCs are presented in the Appendix.

**Discussion**. Theorem 7 and Corollary 2 provide theoretical bounds on the expected loss of a model for a specific demographic group, given the differences in the data distributions between the group and the overall population. These bounds can help explain the empirical results observed in Figure 1. Let's consider the results for AMD detection with ViT (Figure 1a). The overall feature distribution has a mean of 5.92 and a standard deviation of 2.46. For the Asian group, the mean is 6.26 and the standard deviation is 2.53, while for the Black group, the mean is 6.45 and the standard deviation is 2.65. Applying Theorem 7, with probability at least $1 - \delta$, the expected loss for the Asian group can be bounded as follows:

$$\mathbb{E}_{(x,y)\sim\mathcal{D}_{\text{Asian}}}[\ell(f(x),y)] \leq \mathbb{E}_{(x,y)\sim\mathcal{D}}[\ell(f(x),y)] + B|\mu_{\text{Asian}} - \mu|_2 + B\sqrt{|\Sigma_{\text{Asian}} - \Sigma|_F}$$

$$\leq \mathbb{E}_{(x,y)\sim\mathcal{D}}[\ell(f(x),y)] + B|6.26 - 5.92| + B\sqrt{|2.53 - 2.46|^2}$$

$$\leq \mathbb{E}_{(x,y)\sim\mathcal{D}}[\ell(f(x),y)] + 0.34B + 0.07B$$

Similarly, for the Black group:

$$\mathbb{E}_{(x,y)\sim\mathcal{D}_{\text{Black}}}[\ell(f(x),y)] \leq \mathbb{E}_{(x,y)\sim\mathcal{D}}[\ell(f(x),y)] + B|\mu_{\text{Black}} - \mu|_2 + B\sqrt{|\Sigma_{\text{Black}} - \Sigma|_F}$$

$$\leq \mathbb{E}_{(x,y)\sim\mathcal{D}}[\ell(f(x),y)] + B|6.45 - 5.92| + B\sqrt{|2.65 - 2.46|^2}$$

$$\leq \mathbb{E}_{(x,y)\sim\mathcal{D}}[\ell(f(x),y)] + 0.53B + 0.19B$$

These bounds suggest that the expected loss for the Asian and Black groups could be higher than the overall expected loss, which aligns with the lower AUC values observed for these groups in Figure 1a. Corollary 2 further relates the expected loss bound to the distance between the group-specific feature centroids and the overall feature centroid. In Figure 1a, the Asian and Black groups have feature centroids that are farther from the overall centroid compared to the White group, which has a mean of 5.81 and a standard deviation of 2.41. This observation is consistent with the lower AUC values for the Asian and Black groups. Similar patterns can be observed in the other experiments, such as DR detection with ViT (Figure A.2a) and Glaucoma detection with ViT (Figure A.2b). The groups with feature distributions that deviate more from the overall distribution tend to have lower AUC values, as predicted by Theorem 7 and Corollary 2. However, it is important to note that the bounds provided by Theorem 7 and Corollary 2 are probabilistic and depend on the choice of $\delta$. A smaller $\delta$ would lead to a higher probability of the bounds holding but may result in looser bounds. Nevertheless, the theoretical results provide valuable insights into the factors influencing the performance disparities across demographic groups and align well with the empirical observations in Figure 1.

## 5 CONCLUSION

This work presents a comprehensive theoretical analysis of the impact of disease prevalence and data distributions on the fairness guarantees of deep learning models in medical applications. Our analysis reveals the key factors influencing fairness and provides insights into the design and evaluation of equitable algorithms. Future research directions include relaxing distributional assumptions, incorporating additional fairness criteria, developing fairness-aware optimization algorithms, studying the impact of biased data collection and labeling, conducting extensive empirical evaluations, and exploring the fairness-accuracy trade-off. By addressing these open questions, we can continue to advance our understanding of fairness in AI-based medical diagnosis systems and develop more equitable algorithms that ensure fair treatment and outcomes for all patients.

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

# A APPENDIX

## A.1 PROOF OF THEORETICAL ANALYSIS

**Theorem 1** (Connection with Conventional Fairness). *Let $f$ be a classifier, $\ell$ be a loss function bounded by $M$, and $D_{a_i}$ be the data distribution for demographic group $a_i$. Let $p_i$ be the proportion of positive samples in group $a_i$. Then, for any pair of groups $a_i$ and $a_j$:*

$$\left| \mathbb{E}_{(x,y)\sim D_{a_i}}[\ell(f(x),y)] - \mathbb{E}_{(x,y)\sim D_{a_j}}[\ell(f(x),y)] \right| \le \max\{p_i, p_j\} \cdot \left| L^{+,a_i}(f) - L^{+,a_j}(f) \right| + 2M \left| p_i - p_j \right|$$
$$+ \min\{1 - p_i, 1 - p_j\} \cdot \left| L^{-,a_i}(f) - L^{-,a_j}(f) \right|$$

*where $L^{+,a_k}(f) = \mathbb{E}[\ell(f(x),y)|y = 1, s = a_k]$ is the risk of the positive labeled samples in group $a_k$, and $L^{-,a_k}(f) = \mathbb{E}[\ell(f(x),y)|y = 0, s = a_k]$ is the risk of the negative labeled samples in group $a_k$. $L^{+,a_k}(f)$ and $L^{-,a_k}(f)]$ are defined in Donini et al. (2018).*

*Proof.* First, we decompose the expected loss for each group into positive and negative components:

$$\mathbb{E}_{(x,y)\sim D_{a_k}}[\ell(f(x),y)] = p_k \cdot L^{+,a_k}(f) + (1 - p_k) \cdot L^{-,a_k}(f)$$

for $k \in \{i, j\}$.

Now, consider the difference in expected losses:

$$\left| \mathbb{E}_{(x,y)\sim D_{a_i}}[\ell(f(x),y)] - \mathbb{E}_{(x,y)\sim D_{a_j}}[\ell(f(x),y)] \right|$$
$$= \left| (p_i \cdot L^{+,a_i}(f) + (1 - p_i) \cdot L^{-,a_i}(f)) - (p_j \cdot L^{+,a_j}(f) + (1 - p_j) \cdot L^{-,a_j}(f)) \right|$$

Rearrange the terms:

$$= \left| p_i \cdot L^{+,a_i}(f) - p_j \cdot L^{+,a_j}(f) + (1 - p_i) \cdot L^{-,a_i}(f) - (1 - p_j) \cdot L^{-,a_j}(f) \right|$$

Add and subtract $p_i \cdot L^{+,a_j}(f)$ and $(1 - p_j) \cdot L^{-,a_i}(f)$:

$$= \left| p_i \cdot (L^{+,a_i}(f) - L^{+,a_j}(f)) + (p_i - p_j) \cdot L^{+,a_j}(f) \right.$$
$$\left. + (1 - p_j) \cdot (L^{-,a_i}(f) - L^{-,a_j}(f)) + (p_j - p_i) \cdot L^{-,a_i}(f) \right|$$

Apply the triangle inequality:

$$\le p_i \cdot \left| L^{+,a_i}(f) - L^{+,a_j}(f) \right| + |p_i - p_j| \cdot \left| L^{+,a_j}(f) \right|$$
$$+ (1 - p_j) \cdot \left| L^{-,a_i}(f) - L^{-,a_j}(f) \right| + |p_j - p_i| \cdot \left| L^{-,a_i}(f) \right|$$

Note that $p_i \le \max\{p_i, p_j\}$ and $(1 - p_j) \le \min\{1 - p_i, 1 - p_j\}$. Also, since $\ell$ is bounded by $M$, we have $|L^{+,a_j}(f)| \le M$ and $|L^{-,a_i}(f)| \le M$. Apply these bounds:

$$\le \max\{p_i, p_j\} \cdot \left| L^{+,a_i}(f) - L^{+,a_j}(f) \right| + M|p_i - p_j|$$
$$+ \min\{1 - p_i, 1 - p_j\} \cdot \left| L^{-,a_i}(f) - L^{-,a_j}(f) \right| + M|p_j - p_i|$$

We can simplify:

$$\le \max\{p_i, p_j\} \cdot \left| L^{+,a_i}(f) - L^{+,a_j}(f) \right| + 2M \left| p_i - p_j \right|$$
$$+ \min\{1 - p_i, 1 - p_j\} \cdot \left| L^{-,a_i}(f) - L^{-,a_j}(f) \right|$$

This completes the proof. $\qquad\square$

**Theorem 2** (Fairness Error Bound). *Under Assumption 1, given the fairness problem as defined in Definition 1, let $f^*$ be the optimal function that minimizes the maximum absolute difference in the expected loss across all demographic groups. Let $\hat{f}$ be an estimate of $f^*$ based on a finite sample of size $n$. Then, by Hoeffding's inequality, with probability at least $1 - \delta$, the following inequality holds:*

$$\max_{i,j} \left| \mathbb{E}_{(x,y) \sim \mathcal{D}_{a_i}}[\ell(\hat{f}(x), y)] - \mathbb{E}_{(x,y) \sim \mathcal{D}_{a_j}}[\ell(\hat{f}(x), y)] \right| \leq$$

$$\max_{i,j} \left| \mathbb{E}_{(x,y) \sim \mathcal{D}_{a_i}}[\ell(f^*(x), y)] - \mathbb{E}_{(x,y) \sim \mathcal{D}_{a_j}}[\ell(f^*(x), y)] \right| + M\sqrt{\frac{\log(2k/\delta)}{2n \min\{r_i\}}}$$

*Proof.* Let $L_i(f) = \mathbb{E}_{(x,y) \sim \mathcal{D}_{a_i}}[\ell(f(x), y)]$ be the expected loss for demographic group $a_i$ under function $f$. Define $\Delta(f) = \max_{i,j} |L_i(f) - L_j(f)|$ as the maximum absolute difference in the expected loss across all demographic groups.

**Lemma 1** (Hoeffding's Inequality for Bounded Loss). *For any function $f$ and demographic group $a_i$, with probability at least $1 - \delta_i$, the following inequality holds:*

$$|L_i(f) - \hat{L}_i(f)| \leq M\sqrt{\frac{\log(2/\delta_i)}{2n_i}}$$

*where $\hat{L}_i(f) = \frac{1}{n_i} \sum_{j=1}^{n_i} \ell(f(x_j^{(i)}), y_j^{(i)})$ is the empirical loss for group $a_i$, $n_i = nr_i$ is the sample size for group $a_i$, and $(x_j^{(i)}, y_j^{(i)})$ are i.i.d. samples from $\mathcal{D}_{a_i}$.*

The lemma follows from Hoeffding's inequality and the fact that the loss function is bounded. Applying the union bound and setting $\delta_i = \delta/k$ for all $i$, we have that with probability at least $1 - \delta$, the following inequality holds for all demographic groups simultaneously:

$$|L_i(f) - \hat{L}_i(f)| \leq M\sqrt{\frac{\log(2k/\delta)}{2n_i}} \leq M\sqrt{\frac{\log(2k/\delta)}{2n \min\{r_i\}}}$$

Therefore, with probability at least $1 - \delta$, we have:

$$\Delta(\hat{f}) \leq \Delta(f^*) + |\Delta(\hat{f}) - \Delta(f^*)|$$
$$\leq \Delta(f^*) + \max_{i,j} |L_i(\hat{f}) - L_j(\hat{f}) - (L_i(f^*) - L_j(f^*))|$$
$$\leq \Delta(f^*) + \max_{i,j}(|L_i(\hat{f}) - \hat{L}_i(\hat{f})| + |L_j(\hat{f}) - \hat{L}_j(\hat{f})| + |L_i(f^*) - \hat{L}_i(f^*)| + |L_j(f^*) - \hat{L}_j(f^*)|)$$
$$\leq \Delta(f^*) + 4M\sqrt{\frac{\log(2k/\delta)}{2n \min\{r_i\}}}$$

This completes the proof. $\square$

**Theorem 3** (Algorithmic Complexity of Fairness Problem). *Under Assumptions 1 and 2, given the fairness problem as defined in Definition 1 with $k$ demographic groups and a function class $\mathcal{F}$ with finite Vapnik-Chervonenkis (VC) dimension $d$, there exists an algorithm that finds an $\epsilon$-optimal solution $\hat{f}$ to the fairness problem with probability at least $1 - \delta$, using $O(\frac{k^2}{\epsilon^2}(d \log(k/\epsilon) + \log(k/\delta)))$ samples and $O(k^2|\mathcal{F}|)$ time complexity.*

*Proof.* Let $f^* = \arg\min_{f \in \mathcal{F}} \Delta(f)$ be the optimal solution to the fairness problem, where $\Delta(f) = \max_{i,j} |L_i(f) - L_j(f)|$ is the maximum absolute difference in the expected loss across all demographic groups.

**Lemma 2** (Uniform Convergence Bound for Fairness Problem). *Let $\mathcal{F}$ be a function class with finite VC dimension $d$. For any $\epsilon, \delta > 0$, if the sample size $n$ satisfies $n \geq \frac{8M^2k^2}{\epsilon^2}(d \log(16Mk/\epsilon) +$*

$\log(4k^2/\delta))$, *then with probability at least* $1 - \delta$, *the following holds for all* $f \in \mathcal{F}$ *and all pairs of demographic groups* $(i, j)$ *simultaneously:*

$$|L_i(f) - L_j(f) - (\hat{L}_i(f) - \hat{L}_j(f))| \leq \epsilon/2$$

*where* $\hat{L}_i(f) = \frac{1}{n_i} \sum_{j=1}^{n_i} \ell(f(x_j^{(i)}), y_j^{(i)})$ *is the empirical loss for group* $a_i$, *and* $n_i = nr_i$ *is the sample size for group* $a_i$.

Lemma 2 follows from the standard uniform convergence bound for finite VC dimension function classes, applied to the pairwise differences of the loss functions for each pair of demographic groups. Now, consider the following algorithm:

1. Draw a sample of size $n \geq \frac{8M^2k^2}{\epsilon^2}(d\log(16Mk/\epsilon) + \log(4k^2/\delta))$.

2. For each $f \in \mathcal{F}$, calculate the empirical fairness error $\hat{\Delta}(f) = \max_{i,j}|\hat{L}_i(f) - \hat{L}_j(f)|$.

3. Return $\hat{f} = \arg\min_{f \in \mathcal{F}} \hat{\Delta}(f)$.

By Lemma 2, with probability at least $1 - \delta$, we have:

$$\Delta(\hat{f}) \leq \hat{\Delta}(\hat{f}) + \epsilon/2$$
$$\leq \hat{\Delta}(f^*) + \epsilon/2$$
$$\leq \Delta(f^*) + \epsilon$$

Therefore, the algorithm returns an $\epsilon$-optimal solution with probability at least $1 - \delta$. The sample complexity is $O(\frac{k^2}{\epsilon^2}(d\log(k/\epsilon) + \log(k/\delta)))$, and the time complexity is $O(k^2|\mathcal{F}|)$ since we need to calculate the empirical fairness error for each function in $\mathcal{F}$ and each pair of demographic groups. $\square$

**Lemma 3** (Symmetrization). *For any function* $f \in \mathcal{F}$,

$$\mathbb{P}\left[R(f) - R_{emp}(f) > \epsilon\right] \leq 2\mathbb{P}\left[\sup_{f \in \mathcal{F}} |R_{emp}(f) - R'emp(f)| > \frac{\epsilon}{2}\right]$$

*where* $R(f)$ *is the true fairness risk,* $R_{emp}(f)$ *is the empirical fairness risk on the original sample, and* $R'_{emp}(f)$ *is the empirical fairness risk on a ghost sample of size* $m$ *drawn independently from the same distribution as the original sample.*

*Proof.* Let $S = (x_1, y_1), \ldots, (x_m, y_m)$ be the original sample of size $m$ and $S' = (x'_1, y'_1), \ldots, (x'_m, y'_m)$ be the ghost sample of size $m$, both drawn independently from the same distribution. Define the event $A$ as:

$$A = \exists f \in \mathcal{F} : R(f) - R_{emp}(f) > \epsilon$$

We want to bound the probability of event $A$. Consider the following event $B$:

$$B = \left\{\sup_{f \in \mathcal{F}} |R_{emp}(f) - R'_{emp}(f)| > \frac{\epsilon}{2}\right\}$$

We will show that $A \subseteq B \cup B'$, where $B'$ is the same event as $B$ but with the roles of $S$ and $S'$ swapped. Suppose event $A$ occurs, i.e., there exists a function $f \in \mathcal{F}$ such that $R(f) - R_{emp}(f) > \epsilon$. Then, we have:

$$R(f) - R_{emp}(f) > \epsilon \ R(f) - R'emp(f) + R'emp(f) - R_{emp}(f) \quad > \epsilon \ [R(f) - R'emp(f)] + [R'emp(f) - R_{emp}(f)] > \epsilon$$

If $R(f) - R'emp(f) \leq \epsilon/2$, then we must have $R'emp(f) - R_{emp}(f) > \epsilon/2$, which implies event $B$ occurs. On the other hand, if $R(f) - R'emp(f) > \epsilon/2$, then by swapping the roles of $S$ and $S'$, we have $R'emp(f) - R_{emp}(f) > \epsilon/2$, which implies event $B'$ occurs. Therefore, $A \subseteq B \cup B'$, and by the union bound, we have:

$$\mathbb{P}[A] \leq \mathbb{P}[B] + \mathbb{P}[B']$$

Since $S$ and $S'$ are drawn independently from the same distribution, we have $\mathbb{P}[B] = \mathbb{P}[B']$. Thus,

$$\mathbb{P}[A] \leq 2\mathbb{P}[B]$$

which is equivalent to:

$$\mathbb{P}\left[R(f) - R_{\text{emp}}(f) > \epsilon\right] \leq 2\mathbb{P}\left[\sup_{f \in \mathcal{F}} |R_{\text{emp}}(f) - R'_{\text{emp}}(f)| > \frac{\epsilon}{2}\right]$$

This completes the proof. $\qquad\square$

**Theorem 4** (Fairness Generalization Bound). *Under Assumptions 1, given the fairness problem as defined in Definition 1 with $k$ demographic groups and a function space $\mathcal{F}$ with VC dimension $d_{VC}(\mathcal{F})$, for any $\delta > 0$, with probability at least $1 - \delta$, for all $f \in \mathcal{F}$:*

$$\max_{i,j} |R_i(f) - R_j(f)| \leq \max_{i,j} |R_{emp,i}(f) - R_{emp,j}(f)| + M\sqrt{\frac{8(d_{VC}(\mathcal{F})\ln(2em/d_{VC}(\mathcal{F})) + \ln(4k^2/\delta))}{m}}$$

*where $R_i(f) = \mathbb{E}_{(x,y) \sim \mathcal{D}_{a_i}}[\ell(f(x), y)]$ is the expected risk for group $a_i$, $R_{emp,i}(f) = \frac{1}{m_i}\sum_{j=1}^{m_i} \ell(f(x_j^{(i)}), y_j^{(i)})$ is the empirical risk for group $a_i$, $m_i$ is the sample size for group $a_i$, and $m = \sum_{i=1}^{k} m_i$ is the total sample size.*

*Proof.* Let $R(f) = \max_{i,j} |R_i(f) - R_j(f)|$ be the fairness risk and $R_{\text{emp}}(f) = \max_{i,j} |R_{\text{emp},i}(f) - R_{\text{emp},j}(f)|$ be the empirical fairness risk. Based on Lemma 3, by the VC dimension bound in the attached file, we have:

$$\mathbb{P}\left[\sup_{f \in \mathcal{F}} |R_{\text{emp}}(f) - R'_{\text{emp}}(f)| > \frac{\epsilon}{2}\right] \leq 4\Phi(2m)\exp\left(-\frac{m\epsilon^2}{8M^2}\right)$$

where $\Phi(m) = \sum_{i=0}^{d_{\text{VC}}(\mathcal{F})} \binom{m}{i}$ is the growth function of $\mathcal{F}$. Using the bound $\Phi(m) \leq \left(\frac{em}{d_{\text{VC}}(\mathcal{F})}\right)^{d_{\text{VC}}(\mathcal{F})}$ and setting the right-hand side to $\delta/(2k^2)$, we get:

$$\mathbb{P}\left[\sup_{f \in \mathcal{F}} |R_{\text{emp}}(f) - R'_{\text{emp}}(f)| > \frac{\epsilon}{2}\right] \leq \frac{\delta}{2k^2}$$

Applying the union bound over all pairs of demographic groups, we have:

$$\mathbb{P}\left[\exists i, j : R_i(f) - R_j(f) > \epsilon\right] \leq \delta$$

provided that:

$$\epsilon = M\sqrt{\frac{8(d_{\text{VC}}(\mathcal{F})\ln(2em/d_{\text{VC}}(\mathcal{F})) + \ln(4k^2/\delta))}{m}}$$

Therefore, with probability at least $1 - \delta$, for all $f \in \mathcal{F}$:

$$\max_{i,j} |R_i(f) - R_j(f)| \leq \max_{i,j} |R_{\text{emp},i}(f) - R_{\text{emp},j}(f)| + M\sqrt{\frac{8(d_{\text{VC}}(\mathcal{F})\ln(2em/d_{\text{VC}}(\mathcal{F})) + \ln(4k^2/\delta))}{m}}$$

This completes the proof. $\qquad\square$

**Lemma 4** (Uniform Convergence of Fairness Risk). *Under Assumption 2, for any $\delta > 0$, with probability at least $1 - \delta$ over the random choice of $S$:*

$$\sup_{f \in \mathcal{F}} |R(f) - R_{emp}(f, S)| \leq \frac{2LM}{\sqrt{m}}\left(\sqrt{2d_{VC}(\mathcal{F})\ln\frac{em}{d_{VC}(\mathcal{F})}} + \sqrt{2\ln\frac{4}{\delta}}\right)$$

*Proof.* Let $\mathcal{G} = (x, y) \mapsto \ell(f(x), y) : f \in \mathcal{F}$ be the function class induced by the loss function $\ell$ and the function class $\mathcal{F}$. By Assumption 2, the loss function $\ell$ is bounded by $M$ and Lipschitz continuous with constant $L$. For any function $g \in \mathcal{G}$, we have:

$$|g(x, y)| = |\ell(f(x), y)| \leq M$$

and for any $(x_1, y_1), (x_2, y_2)$,

$$|g(x_1, y_1) - g(x_2, y_2)| = |\ell(f(x_1), y_1) - \ell(f(x_2), y_2)| \quad \leq L|f(x_1) - f(x_2)| \leq LD|x_1 - x_2|$$

where $D$ is the Lipschitz constant of functions in $\mathcal{F}$. Therefore, functions in $\mathcal{G}$ are bounded by $M$ and Lipschitz continuous with constant $LD$. By McDiarmid's inequality, for any $f \in \mathcal{F}$, with probability at least $1 - \delta/2$,

$$|R(f) - R_{\text{emp}}(f, S)| \leq M\sqrt{\frac{2\ln(2/\delta)}{m}}$$

Let $N(\epsilon, \mathcal{F}, |\cdot|\infty)$ be the $\epsilon$-covering number of $\mathcal{F}$ with respect to the $L\infty$ norm. By the Lipschitz continuity of the loss function, we have:

$$N(\epsilon, \mathcal{G}, |\cdot|\infty) \leq N(\epsilon/L, \mathcal{F}, |\cdot|\infty)$$

By the VC dimension bound on the covering number Van Der Vaart & Wellner (1997), we have:

$$N(\epsilon/L, \mathcal{F}, |\cdot|\infty) \leq \left(\frac{2eL}{\epsilon}\right)^{d_{\text{VC}}(\mathcal{F})}$$

By the union bound and the covering number bound, with probability at least $1 - \delta/2$, for all $f \in \mathcal{F}$,

$$|R(f) - R_{\text{emp}}(f, S)| \leq M\sqrt{\frac{2(d_{\text{VC}}(\mathcal{F})\ln(2eL/\epsilon) + \ln(2/\delta))}{m}}$$

Setting $\epsilon = LM\sqrt{\frac{2d_{\text{VC}}(\mathcal{F})\ln(em/d_{\text{VC}}(\mathcal{F}))}{m}}$, we get:

$$\sup_{f \in \mathcal{F}} |R(f) - R_{\text{emp}}(f, S)| \leq \frac{2LM}{\sqrt{m}}\left(\sqrt{2d_{\text{VC}}(\mathcal{F})\ln\frac{em}{d_{\text{VC}}(\mathcal{F})}} + \sqrt{2\ln\frac{4}{\delta}}\right)$$

with probability at least $1 - \delta$ over the random choice of $S$. This completes the proof. $\qquad\square$

**Theorem 5** (Convergence of Fairness Risk Minimizer). *Let $\mathcal{F}$ be a function space with VC dimension $d_{VC}(\mathcal{F})$ and let $f^* = \arg\min_{f \in \mathcal{F}} R(f)$ be the fairness risk minimizer. Let $\hat{f}_S = \arg\min_{f \in \mathcal{F}} R_{emp}(f, S)$ be the empirical fairness risk minimizer based on a training set $S$ of size $m$. Then, under Assumptions 1 and 2, for any $\delta > 0$, with probability at least $1 - \delta$ over the random choice of $S$:*

$$R(\hat{f}_S) - R(f^*) \leq \frac{2LM}{\sqrt{m}}\left(\sqrt{2dVC(\mathcal{F})\ln\frac{em}{d_{VC}(\mathcal{F})}} + \sqrt{2\ln\frac{4}{\delta}}\right)$$

*Proof.* The proof relies on Lemma 4. Let $\epsilon = \frac{2LM}{\sqrt{m}}\left(\sqrt{2d_{\text{VC}}(\mathcal{F})\ln\frac{em}{d_{\text{vc}}(\mathcal{F})}} + \sqrt{2\ln\frac{4}{\delta}}\right)$. By the lemma, with probability at least $1 - \delta$ over the random choice of $S$:

$$R(\hat{f}_S) \leq R_{\text{emp}}(\hat{f}_S, S) + \epsilon \qquad\qquad \leq R_{\text{emp}}(f^, S) + \epsilon \leq R(f^) + 2\epsilon$$

Therefore, with probability at least $1 - \delta$ over the random choice of $S$:

$$R(\hat{f}_S) - R(f^*) \leq \frac{2LM}{\sqrt{m}}\left(\sqrt{2d\text{VC}(\mathcal{F})\ln\frac{em}{d_{\text{VC}}(\mathcal{F})}} + \sqrt{2\ln\frac{4}{\delta}}\right)$$

This completes the proof. $\qquad\square$

**Lemma 5** (Risk Bound for Normal Distributions). *Let $p_i$ and $p$ be the density functions of two normal distributions with means $\mu_i$ and $\mu$, and covariance matrices $\Sigma_i$ and $\Sigma$, respectively. Then, for any function $f$:*

$$\left|\mathbb{E}_{(x,y)\sim p_i}[f(x,y)] - \mathbb{E}_{(x,y)\sim p}[f(x,y)]\right| \leq L_f|\mu_i - \mu|_2 + L_f\sqrt{|\Sigma_i - \Sigma|_F}$$

*where $L_f$ is the Lipschitz constant of $f$ with respect to the Euclidean norm.*

*Proof.* Let $\mathcal{P}(\mathbb{R}^d \times \mathbb{R})$ be the set of all probability measures on $\mathbb{R}^d \times \mathbb{R}$, and let $\Pi(p_i, p)$ be the set of all joint probability measures on $(\mathbb{R}^d \times \mathbb{R}) \times (\mathbb{R}^d \times \mathbb{R})$ with marginals $p_i$ and $p$. The Kantorovich-Rubinstein duality states that for any Lipschitz function $f$ with Lipschitz constant $L_f$, we have:

$$|\mathbb{E}(x,y) \sim p_i[f(x,y)] - \mathbb{E}(x,y) \sim p[f(x,y)]| \le L_f \cdot W_1(p_i, p) \tag{2}$$

where $W_1(p_i, p)$ is the 1-Wasserstein distance between $p_i$ and $p$, defined as:

$$W_1(p_i, p) = \inf_{\gamma \in \Pi(p_i, p)} \mathbb{E}_{((x,y),(x',y')) \sim \gamma}[|(x,y) - (x',y')|_2] \tag{3}$$

Now, let's focus on bounding the 1-Wasserstein distance between two normal distributions $p_i = \mathcal{N}(\mu_i, \Sigma_i)$ and $p = \mathcal{N}(\mu, \Sigma)$. By the triangular inequality, we have:

$$W_1(p_i, p) \le \inf_{\gamma \in \Pi(p_i, p)} \mathbb{E}((x,y), (x',y')) \sim \gamma[|x - x'|2] + \inf \gamma \in \Pi(p_i, p)\mathbb{E}((x,y), (x',y')) \sim \gamma[|y - y'|_2] \tag{4}$$

$$\le W_1(\mathcal{N}(\mu_i, \Sigma_i), \mathcal{N}(\mu, \Sigma_i)) + W_1(\mathcal{N}(\mu, \Sigma_i), \mathcal{N}(\mu, \Sigma)) \tag{5}$$

The first term in equation 5 can be bounded by the mean difference:

$$W_1(\mathcal{N}(\mu_i, \Sigma_i), \mathcal{N}(\mu, \Sigma_i)) \le |\mu_i - \mu|_2 \tag{6}$$

The second term in equation 5 can be bounded by the covariance difference Givens & Shortt (1984):

$$W_1(\mathcal{N}(\mu, \Sigma_i), \mathcal{N}(\mu, \Sigma)) \le \sqrt{|\Sigma_i - \Sigma|_F} \tag{7}$$

Combining equation 2, equation 5, equation 6, and equation 7, we obtain:

$$|\mathbb{E}(x,y) \sim p_i[f(x,y)] - \mathbb{E}(x,y) \sim p[f(x,y)]| \le L_f|\mu_i - \mu|_2 + L_f\sqrt{|\Sigma_i - \Sigma|_F}$$

which completes the proof. $\qquad\square$

**Theorem 6** (Group-Specific Risk Bound Theorem for Normal Distributions). *Let $\mathcal{F}$ be a function space with VC dimension $d_{VC}(\mathcal{F})$ and let $f_i^* = \arg\min_{f \in \mathcal{F}} R_i(f)$ be the risk minimizer for demographic group $a_i$. Let $\hat{f}_S = \arg\min_{f \in \mathcal{F}} R_{emp}(f, S)$ be the empirical risk minimizer based on a training set $S$ of size $m$ drawn from the overall data distribution. Then, under Assumptions 2 and 3, for any $\delta > 0$, with probability at least $1 - \delta$ over the random choice of $S$:*

$$R_i(\hat{f}_S) - R_i(f_i^*) \le \frac{2LM}{\sqrt{m}}\left(\sqrt{2dVC(\mathcal{F})\ln\frac{em}{dVC(\mathcal{F})}} + \sqrt{2\ln\frac{4}{\delta}}\right) + L|\mu_i - \mu|_2 + L\sqrt{|\Sigma_i - \Sigma|F}$$

*where $R_i(f) = \mathbb{E}_{(x,y) \sim \mathcal{D}_{a_i}}[\ell(f(x),y)]$ is the expected risk for group $a_i$, $R_{emp}(f, S) = \frac{1}{m}\sum_{j=1}^m \ell(f(x_j), y_j)$ is the empirical risk based on the training set $S$, $\mu$ and $\Sigma$ are the mean and covariance matrix of the overall data distribution, and $|\cdot|_F$ denotes the Frobenius norm.*

*Proof.* The proof relies on Lemma 5. Let $\epsilon = \frac{2LM}{\sqrt{m}}\left(\sqrt{2d_{VC}(\mathcal{F})\ln\frac{em}{d_{VC}(\mathcal{F})}} + \sqrt{2\ln\frac{4}{\delta}}\right)$. By Lemma 4, with probability at least $1 - \delta$ over the random choice of $S$::

$$R_{emp}(\hat{f}_S, S) - R(\hat{f}_S) \le \epsilon, \quad R_{emp}(f_i^*, S) - R(f_i^*) \ge -\epsilon$$

Applying Lemma 5 with $f(x,y) = \ell(\hat{y}, y)$, which has Lipschitz constant $L$ by Assumption 2, we have:

$$R_i(f) - R(f) \le L|\mu_i - \mu|_2 + L\sqrt{|\Sigma_i - \Sigma|_F}, \quad \forall f \in \mathcal{F}$$

Combining the above inequalities, with probability at least $1 - \delta$ over the random choice of $S$:

$$\begin{aligned}
R_i(\hat{f}_S) - R_i(f_i^*) &= R_i(\hat{f}_S) - R(\hat{f}_S) + R(\hat{f}_S) - R_{emp}(\hat{f}_S, S) \\
&\quad + R_{emp}(\hat{f}_S, S) - R_{emp}(f_i^*, S) + R_{emp}(f_i^*, S) - R(f_i^*) \\
&\quad + R(f_i^*) - R_i(f_i^*) \\
&\le L|\mu_i - \mu|_2 + L\sqrt{|\Sigma_i - \Sigma|_F} + \epsilon + 0 + \epsilon \\
&\quad + L|\mu_i - \mu|2 + L\sqrt{|\Sigma_i - \Sigma|F} \\
&= \frac{2LM}{\sqrt{m}}\left(\sqrt{2d\text{VC}(\mathcal{F})\ln\frac{em}{d\text{VC}(\mathcal{F})}} + \sqrt{2\ln\frac{4}{\delta}}\right) \\
&\quad + 2L|\mu_i - \mu|_2 + 2L\sqrt{|\Sigma_i - \Sigma|_F}
\end{aligned}$$

This completes the proof. $\qquad\square$

**Theorem 7** (Expected Loss Bound for Demographic Group with Normal Distribution). *Let $\mathcal{D}_{a_i} \sim \mathcal{N}(\mu_i, \Sigma_i)$ be the data distribution for demographic group $a_i$, and let $\mathcal{D} \sim \mathcal{N}(\mu, \Sigma)$ be the overall data distribution. Let $f(\cdot)$ be a function that maps input $x$ to predicted labels $\hat{y}$, and let $\ell$ be a loss function bounded by $B$, i.e., $|\ell(f(x), y)| \leq B$ for all $x, y$. Suppose we have a training set $(x_j, y_j)_{j=1}^n$ of size $n$ drawn independently from $\mathcal{D}$. Then, with probability at least $1 - \delta$, the expected loss of $f(\cdot)$ on $\mathcal{D}_{a_i}$ is bounded as follows:*

$$\mathbb{E}_{(x,y)\sim\mathcal{D}_{a_i}}[\ell(f(x),y)] \leq \mathbb{E}_{(x,y)\sim\mathcal{D}}[\ell(f(x),y)] + B|\mu_i - \mu|_2 + B\sqrt{|\Sigma_i - \Sigma|_F}$$

*Proof.* We start by applying the reference theorem to the overall data distribution $\mathcal{D}$:

$$\mathbb{E}_{(x,y)\sim\mathcal{D}}[\ell(f(x),y)] \leq \frac{1}{n}\sum_{j=1}^n \ell(f(x_j), y_j) + B\sqrt{\frac{2|\Sigma|\log(2/\delta)}{n}}$$

Now, we use Lemma 5 to relate the expected loss on $\mathcal{D}_{a_i}$ to the expected loss on $\mathcal{D}$: Applying the lemma with $f(x,y) = \ell(f(x),y)$, which has Lipschitz constant $B$ since $\ell$ is bounded by $B$, we have:

$$\left|\mathbb{E}_{(x,y)\sim\mathcal{D}_{a_i}}[\ell(f(x),y)] - \mathbb{E}_{(x,y)\sim\mathcal{D}}[\ell(f(x),y)]\right| \leq B|\mu_i - \mu|_2 + B\sqrt{|\Sigma_i - \Sigma|_F}$$

Rearranging the terms, we get:

$$\mathbb{E}_{(x,y)\sim\mathcal{D}_{a_i}}[\ell(f(x),y)] \leq \mathbb{E}_{(x,y)\sim\mathcal{D}}[\ell(f(x),y)] + B|\mu_i - \mu|_2 + B\sqrt{|\Sigma_i - \Sigma|_F}$$

This completes the proof. $\square$

**Theorem 7** (Expected Loss Bound for Demographic Group with Normal Distribution). *Let $\mathcal{D}_{a_i} \sim \mathcal{N}(\mu_i, \Sigma_i)$ be the data distribution for demographic group $a_i$, and let $\mathcal{D} \sim \mathcal{N}(\mu, \Sigma)$ be the overall data distribution. Let $f(\cdot)$ be a function that maps input $x$ to predicted labels $\hat{y}$, and let $\ell$ be a loss function bounded by $B$, i.e., $|\ell(f(x), y)| \leq B$ for all $x, y$. Suppose we have a training set $(x_j, y_j)_{j=1}^n$ of size $n$ drawn independently from $\mathcal{D}$. Then, with probability at least $1 - \delta$, the expected loss of $f(\cdot)$ on $\mathcal{D}_{a_i}$ is bounded as follows:*

$$\mathbb{E}_{(x,y)\sim\mathcal{D}_{a_i}}[\ell(f(x),y)] \leq \mathbb{E}_{(x,y)\sim\mathcal{D}}[\ell(f(x),y)] + B|\mu_i - \mu|_2 + B\sqrt{|\Sigma_i - \Sigma|_F}$$

*Proof.* By the triangle inequality and Theorems 5 and 6, we have:

$$R_i(f^*) - R_i(f_i^*) \leq R_i(f^*) - R_i(\hat{f}_S) + R_i(\hat{f}_S) - R_i(f_i^*)$$

$$\leq \frac{2LM}{\sqrt{m}}\sqrt{2d_{VC}(\mathcal{F})\ln\frac{em}{d_{VC}(\mathcal{F})} + 2\ln\frac{4}{\delta}}$$

$$+ \frac{2LM}{\sqrt{m}}\sqrt{2d_{VC}(\mathcal{F})\ln\frac{em}{d_{VC}(\mathcal{F})} + 2\ln\frac{4}{\delta}} + L|\mu_i - \mu|2 + L\sqrt{|\Sigma_i - \Sigma|F}$$

$$= \frac{4LM}{\sqrt{m}}\sqrt{2dVC(\mathcal{F})\ln\frac{em}{dVC(\mathcal{F})} + 2\ln\frac{4}{\delta}} + L|\mu_i - \mu|_2 + L\sqrt{|\Sigma_i - \Sigma|_F}$$

This completes the proof. $\square$

**Corollary 2** (Correlation between Expected Loss Bound and Feature Distance). *Let $\mathcal{D}_{a_i} \sim \mathcal{N}(\mu_i, \Sigma_i)$ be the data distribution for demographic group $a_i$, and let $\mathcal{D} \sim \mathcal{N}(\mu, \Sigma)$ be the overall data distribution. Let $f(\cdot)$ be a function that maps input $x$ to a discriminative feature $z$ in a metric space, and let $\ell$ be a loss function bounded by $B$, i.e., $|\ell(z, y)| \leq B$ for all $z, y$. Suppose we have a training set $(x_j, y_j)j = 1^n$ of size $n$ drawn independently from $\mathcal{D}$. Let $\bar{z}$ be the centroid of the features generated by $f$ on the overall data distribution, and let $\bar{z}_i$ be the centroid of the features generated by $f$ on the demographic group $a_i$. Then, with probability at least $1 - \delta$, the expected loss of $f(\cdot)$ on $\mathcal{D}_{a_i}$ is bounded as follows:*

$$\mathbb{E}_{(x,y)\sim\mathcal{D}_{a_i}}[\ell(f(x),y)] \leq \mathbb{E}_{(x,y)\sim\mathcal{D}}[\ell(f(x),y)] + B \cdot d(\bar{z}_i, \bar{z}) + B\sqrt{\mathbb{E}_{z\sim f(\mathcal{D}_{a_i})}[d^2(z, \bar{z}_i)] - \mathbb{E}_{z\sim f(\mathcal{D})}[d^2(z, \bar{z})]}$$

*where $d(\cdot, \cdot)$ is a distance function in the metric space.*

*Proof.* We start by applying Theorem 7 to the discriminative feature space:

$$\mathbb{E}_{(x,y)\sim\mathcal{D}_{a_i}}[\ell(f(x),y)] \leq \mathbb{E}_{(x,y)\sim\mathcal{D}}[\ell(f(x),y)] + B|\mu z_i - \mu_z|2 + B\sqrt{|\Sigma z_i - \Sigma_z|F}$$

where $\mu z_i$ and $\Sigma_{z_i}$ are the mean and covariance matrix of the features generated by $f$ on $\mathcal{D}_{a_i}$, and $\mu_z$ and $\Sigma_z$ are the mean and covariance matrix of the features generated by $f$ on $\mathcal{D}$. By the definition of the centroid, we have $\bar{z}_i = \mu z_i$ and $\bar{z} = \mu_z$. Therefore, $|\mu_{z_i} - \mu_z|_2 = d(\bar{z}_i, \bar{z})$. Now, let's focus on the term $|\Sigma_{z_i} - \Sigma_z|_F$. By the properties of the Frobenius norm and the trace operator Zhang (2011), we have:

$$|\Sigma_{z_i} - \Sigma_z|F = \sqrt{\text{tr}((\Sigma z_i - \Sigma_z)^T(\Sigma_{z_i} - \Sigma_z))}$$

$$= \sqrt{\text{tr}(\Sigma_{z_i}^2) - 2\text{tr}(\Sigma_{z_i}\Sigma_z) + \text{tr}(\Sigma_z^2)}$$

$$= \sqrt{\mathbb{E}_{z\sim f(\mathcal{D}_{a_i})}[d^2(z,\bar{z}_i)] - 2\mathbb{E}_{z\sim f(\mathcal{D}_{a_i}),z'\sim f(\mathcal{D})}[d(z,z')] + \mathbb{E}_{z\sim f(\mathcal{D})}[d^2(z,\bar{z})]}$$

Using the Cauchy-Schwarz inequality, we can bound the cross-term as follows:

$$\mathbb{E}_{z\sim f(\mathcal{D}_{a_i}),z'\sim f(\mathcal{D})}[d(z,z')] \leq \sqrt{\mathbb{E}_{z\sim f(\mathcal{D}_{a_i})}[d^2(z,\bar{z}_i)] \cdot \mathbb{E}_{z\sim f(\mathcal{D})}[d^2(z,\bar{z})]}$$

$$\leq \frac{1}{2}\left(\mathbb{E}_{z\sim f(\mathcal{D}_{a_i})}[d^2(z,\bar{z}_i)] + \mathbb{E}_{z\sim f(\mathcal{D})}[d^2(z,\bar{z})]\right)$$

Substituting this bound into the expression for $|\Sigma_{z_i} - \Sigma_z|_F$, we get:

$$|\Sigma_{z_i} - \Sigma_z|F \leq \sqrt{\mathbb{E}_{z\sim f(\mathcal{D}_{a_i})}[d^2(z,\bar{z}_i)] - \mathbb{E}_{z\sim f(\mathcal{D})}[d^2(z,\bar{z})]}$$

Combining the results for $|\mu_{z_i} - \mu_z|2$ and $|\Sigma z_i - \Sigma_z|_F$, we obtain the desired bound:

$$\mathbb{E}_{(x,y)\sim\mathcal{D}_{a_i}}[\ell(f(x),y)] \leq \mathbb{E}_{(x,y)\sim\mathcal{D}}[\ell(f(x),y)] + B\cdot d(\bar{z}_i,\bar{z}) + B\sqrt{\mathbb{E}_{z\sim f(\mathcal{D}_{a_i})}[d^2(z,\bar{z}_i)] - \mathbb{E}_{z\sim f(\mathcal{D})}[d^2(z,\bar{z})]}$$

This completes the proof. $\square$

### A.2 RESULTS ON HAM10000

Figure A.1 shows the feature distribution, AUC, and Brier Score with EfficientNet for Basal Cell Carcinoma (BCC) detection on the HAM10000 dataset across three attributes: Gender and Age. It can be observed that the overall feature distribution has a mean of 12.39 and a standard deviation of 4.14. For the female group, the mean is 12.25 with a standard deviation of 4.09, while for the male group, the mean is 12.52 with a standard deviation of 4.17. By comparison, it can be found that groups with feature distributions deviating more from the overall distribution tend to have lower AUC values, which aligns with the predictions of Theorem 7 and Corollary 2.

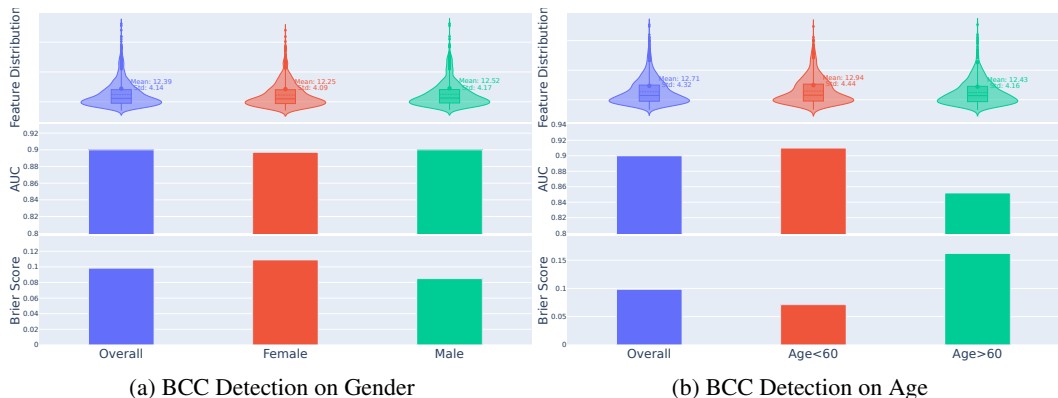

(a) BCC Detection on Gender        (b) BCC Detection on Age

Figure A.1: Combination of feature distribution, AUC, and Brier Score with EfficientNet for Basal Cell Carcinoma (BCC) detection on the **HAM10000 dataset** across two attributes: Gender and Age.

### A.3 RESULTS ON FAIRFACE

Figure A.2 illustrates the feature distribution, AUC, and Brier Score of EfficientNet for glasses detection using the FairFace dataset, evaluated across three different attributes: Age, Skin Color, and Gender. Taking the age group analysis as an example, the overall feature distribution exhibits a mean of 10.21 and a standard deviation of 2.45. Specifically, for the age¡35 group, the mean is 9.88 with a standard deviation of 1.87; for the 35¡age¡65 group, the mean is 10.31 with a standard deviation of 2.56; and for the age¿65 group, the mean is 10.29 with a standard deviation of 2.74. A comparative evaluation shows that demographic groups whose feature distributions deviate more significantly from the overall distribution tend to achieve lower AUC values. This observation is in line with the theoretical predictions derived from Theorem 7 and Corollary 2.

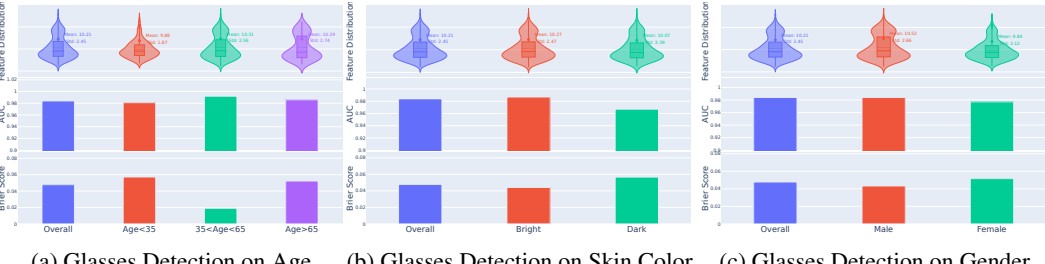

(a) Glasses Detection on Age    (b) Glasses Detection on Skin Color    (c) Glasses Detection on Gender

Figure A.2: Combination of feature distribution, AUC, and Brier Score with EfficientNet for Glasses detection on the **FairFace dataset** across three attributes: Age, Skin Color, and Gender.

