# OpenReview forum: "Impact of Data Distribution on Fairness Guarantees in Equitable Deep Learning"
_ICLR.cc/2025/Conference — ICLR 2025 Conference Withdrawn Submission_

### Official Review · Reviewer_cpA1 · 2024-10-29

**Soundness:** 3
**Presentation:** 3
**Contribution:** 2
**Rating:** 6
**Confidence:** 4

**Summary:**

This work presents an interesting learning-theory-inspired analysis of fair machine learning, focussing on fairness measures that aim to equalize the loss across all groups. Specifically, the fairness measure is of a similar spirit as equalized odds/equal opportunity and is defined as the differences in expected loss across various demographic groups. Next, they derive interesting complexity bounds on this loss difference using statistical learning techniques like Hoeffding's inequality, VC dimension, and the symmetrization lemma. They have also included experimental results on several datasets alongside their theoretical contributions.

**Strengths:**

-- Interesting mathematical analysis using techniques from learning theory such as Hoeffding bound, VC dimension, and Symmetrization lemma.
-- The paper is generally well-written and the ideas are quite nicely presented.
-- They have also included experimental results to show how their upper bound can be computable in several scenarios. The main strength is that the theoretical bound seems to be computable from the experiments, so the research has both depth and applicability.

**Weaknesses:**

-- Though they mention AI-based medical diagnosis here and there including the abstract, I don't think the paper has anything unique to medical diagnosis here. I think the emphasis on medical diagnosis is a bit of a distraction and can be discussed only in experiments if needed.

 -- While the derivation of generalization bounds for a loss function in itself is not new, their main nuance (as per my understanding) lies in bounding the difference of the losses. They also make Lipschitz assumptions. I can increase my rating if the novelty of the analysis is distilled out.

-- The problem statement is closer to accuracy-fairness tradeoffs. While the paper has referenced several early papers in this area of accuracy-fairness tradeoffs, a lot of other prior works in the last 2-3 years that are quite closely related to this work have not been discussed.
[1] Menon, A. K. and Williamson, R. C. The cost of fairness in binary classification. In Proceedings of the Conference on
Fairness, Accountability and Transparency, 2018.
[2] Zhao, H. and Gordon, G. J. Inherent tradeoffs in learning fair representation. arXiv preprint arXiv:1906.08386, 2019.
[3] Sanghamitra Dutta, Dennis Wei, Hazar Yueksel, Pin-Yu Chen, Sijia Liu, Kush Varshney. Is There a Trade-Off Between Fairness and Accuracy? A Perspective Using Mismatched Hypothesis Testing. International Conference on Machine Learning 2020.
[4] Garg, S., Kim, M. P., and Reingold, O. Tracking and improving information in the service of fairness. In Proceedings
of the ACM Conference on Economics and Computation, pp. 809–824, 2019.

For instance, [1] also considers similar fairness metrics. [3] also looks into tradeoffs using equalized-odds-like measures and difference in errors across groups.

-- I would also be curious if this type of analysis has been explored in the context of fairness in federated learning attempting to characterize the worst gap in loss across multiple clients.

-- Another possible limitation: It might be difficult to extend this to demographic parity?

**Questions:**

-- Could you highlight the main steps or nuances in the mathematical analysis that arises due to difference of loss in comparison to standard generalization bounds on loss functions?

-- In the experiments section, are you comparing an upper bound with an upper bound?

-- What would be the main takeaway of the experiments section?

---

### Official Review · Reviewer_rrPY · 2024-11-02

**Soundness:** 2
**Presentation:** 2
**Contribution:** 2
**Rating:** 3
**Confidence:** 3

**Summary:**

The paper presents theoretical results regarding the fairness losses of machine learning models. These theoretical results are then validated on different medical datasets.

**Strengths:**

1. The paper derives a range of theoretical results involving fairness error bounds, algorithmic complexity, generalization bounds, convergence rates, and group-specific risk bounds
2. The paper also conducts extensive experiments on a variety of medical datasets to confirm the theoretical findings.

**Weaknesses:**

1. Firstly, the authors frame the problem as specifically for AI-based medical diagnosis systems. This is also reflected by the mention of specifically the medical setting in both, the abstract and the introduction. However, the setting being considered is much more general, and therefore should not be framed as being specific to the medical setting.
2. The motivation behind the paper is not very clear. The authors derive a number of theoretical results, however, these results are not well motivated. For example, it is not clear how these results can be useful in practice, or how they can help improve fair model training.
3. There is no discussion of the implications of theorem 1. Why is it useful and what insights does it provide?
4. The disease prevalence $r_i$ is not defined formally before assumption 1.
5. In Theorem 2, what is the loss that the optimal function f^* minimises?
6. I am not convinced that the result in Theorem 2 is correct. Firstly, there is no assumption on how close $\hat{f}$ is to $f^*$. So in theory, $\hat{f}$ could be very ‘far away’ from $f^*$ if it is not trained correctly. In this case, even as the number of data $n$ increases, the fairness errors of model $\hat{f}$ could be very different from that of $f^*$. In specific, in the proof of this result, how do you get from line 790 to line 791 (i.e. from the second inequality to the third)?
7. $\epsilon$-optimality is not defined
8. > The theorem suggests that to achieve a smaller fairness risk, one should have a larger sample size, a smaller VC dimension, and a smaller number of demographic groups (lines 263-265)

This is not necessarily true. This just means that the upper bound is small in this case, but does not necessarily mean that these parameters lead to a smaller fairness risk

9. fairness risk in line 273 $R(f)$ is not defined explicitly.
10. There is no discussion on how realistic the assumptions made are, and how robust the theoretical and empirical results are to these assumptions.

**Questions:**

> We prove that under certain conditions, the local optima of the fairness problem can outperform those of the supervised learning problem, highlighting the importance of considering fairness criteria in model development.

Can you please elaborate what this means?

Please see the weakness section above.

---

### Official Review · Reviewer_JtZC · 2024-11-07

**Soundness:** 3
**Presentation:** 4
**Contribution:** 3
**Rating:** 5
**Confidence:** 2

**Summary:**

This paper presents a theoretical framework for analyzing fairness in medical domains across diverse demographic groups. The authors present several strong analytical results, under some statistical assumptions. The authors evaluate on 4 datasets on two deep learning models with fairness over racial groups.

**Strengths:**

1. Overall, the evaluation seems reasonable within the specific domain. The authors present 4 real-world datasets for different detection tasks.

2. The analytical results seem quite strong. Specifically Thm 7 and Cor 2 could be quite useful results for analyzing fairness under Gaussian assumptions.

**Weaknesses:**

1. Overall, the evaluation in this work is quite weak. The primary result, Fig 1 is still a mystery to me. I dont have intuition for what the feature distribution 'ought' look like. So this seems the authors present mostly AUC over 4 detection tasks.

Unless I missed it, this work doesn't actually present any bias mitigation strategy, except some discussion about sufficiently large sampling.

2. there seems to be some assumptions of normality within this work that might not

3. The overall scope of this work is somewhat limited. I didn't quite get the specifics that make these bounds hold under *medical domains* specifically (vs. domain independent). Why this is a domain paper is still a mystery to me, as none of the problem setting particularize it to medical.

Small: While the high level analytical results are fairly intuitive. I did seem to get lost in the theorem specifics. This could be an issue with me, or the notation, which I often found impenetrable without further highlighting or description in text. e.g. Thm 1, 6. Corr. 1,2.

Overall, in my (unconfident) estimation, this could be a reasonable domain paper with strong analytical results. However, without a mitigation strategy, with some difficult interpreting the theorems, and without understanding the domain specificity (thus narrowing the paper), I'm not over the accept threshold on this.

**Questions:**

1. I actually don't see that the main results (like Thm 7) have the sample size as a factor in the bound? Is this correct? Shouldn't the bound improve as a factor of n?

---

### Note · Authors · 2024-11-15

I have read and agree with the venue's withdrawal policy on behalf of myself and my co-authors.